# Design and Evaluation of Composite Magnetic Iron–Platinum Nanowires for Targeted Cancer Nanomedicine

**DOI:** 10.3390/biomedicines11071857

**Published:** 2023-06-29

**Authors:** Abu Bakr Nana, Thashree Marimuthu, Daniel Wamwangi, Pierre P. D. Kondiah, Yahya E. Choonara

**Affiliations:** 1Wits Advanced Drug Delivery Platform Research Unit, Department of Pharmacy and Pharmacology, School of Therapeutic Sciences, Faculty of Health Sciences, University of the Witwatersrand, 7 York Road, Parktown, Johannesburg 2193, South Africa; abubakr.nana@gmail.com (A.B.N.); thashree.marimuthu@wits.ac.za (T.M.); pierre.kondiah@wits.ac.za (P.P.D.K.); 2School of Physics, Materials Physics Research Institute, University of the Witwatersrand, Private Bag 3, WITS, Johannesburg 2050, South Africa; daniel.wamwangi@wits.ac.za

**Keywords:** cancer, magnetic nanowires, magnetic targeted delivery, cell internalization, template electrodeposition

## Abstract

The purpose of the study was to synthesize and investigate the influence of geometrical structure, magnetism, and cytotoxic activity on core–shell platinum and iron–platinum (Fe/Pt) composite nanowires (NWs) for potential application in targeted chemotherapeutic approaches. The Pt-NWs and Fe/Pt composite NWs were synthesized via template electrodeposition, using anodic aluminum oxide (AAO) membranes. The Fe/Pt composite NWs (Method 1) was synthesized using two electrodeposition steps, allowing for greater control of the diameter of the NW core. The Fe/Pt composite NWs (Method 2) was synthesized by pulsed electrodeposition, using a single electrolytic bath. The properties of the synthesized NWs were assessed by high-resolution transmission electron microscopy (HRTEM), Raman spectroscopy, powder X-ray diffraction (XRD), inductively coupled plasma–optical emission spectrometry (ICP-OES), vibrating-sample magnetometry (VSM), and surface charge (zeta potential). A microscopy image analysis of the NWs revealed the presence of high-aspect-ratio NWs with nominal diameters of 40–50 nm and lengths of approximately <4 µm. The obtained powder XRD patterns confirmed the presence of a polycrystalline structure for both Pt NWs and Fe/Pt composite NWs. The potential utility of the synthesized NW nanoplatforms for anticancer activity was investigated using Tera 1 cells and Mouse 3T3 cells. Pt-NWs displayed modest cytotoxic activity against Tera 1 cells, while the Fe/Pt composite NWs (both Methods 1 and 2) demonstrated enhanced cytotoxic activity compared to the Pt-NWs on Tera 1 cells. The Fe/Pt composite NWs (Method 1) displayed ferromagnetic behavior and enhanced cytotoxic activity compared to Pt-NWs on Tera 1 cells, thus providing a sound basis for future magnetically targeted chemotherapeutic applications.

## 1. Introduction

Despite the advances in cancer therapeutics, high mortality and incidence rates prevail with certain types of cancers [1,2]. The majority of chemotherapeutic agents widely bio-distribute, leading to severe and debilitating side effects [3]. Hence, alternate strategies that minimize these adverse effects are required to contribute to alleviating these limitations of conventional chemotherapy.

For example, magnetic oxides and their composites have been extensively explored to provide more targeted delivery of anticancer drugs against specific cancers. In addition, the use of composite nanostructures of varying shape and bioactivity allows for the summation of the individual critical biomaterial properties to achieve an optimal performance for chemotherapy.

Core–shell nanowires (NWs), for instance, are cylindrical nanostructures with high aspect ratios and shape anisotropy that intrinsically allow for greater coercivity and high magnetic moments per volume compared to spherical nanostructures [4,5]. In the search for alternative nanoplatforms to deliver chemotherapy in specific cancers, NWs can provide a promising alternative to the suite of nanomedicines for targeted chemotherapy. Coupled with magnetic oxides such as Fe and FeO, NWs can exhibit bio-(soft) ferromagnetic behavior, and their magnetic properties can be modulated by manipulating the thickness of the oxide shell layer formed and crystallographic orientation during template electrodeposition [6,7].

In addition, bio-metals such as platinum (Pt)-based nanoparticles (NPs) have demonstrated significant potential in nanomedicine [8], radiation dose enhancement [9], and photothermal therapy [10]. The progression of Pt-NPs to anticancer activity is promising based on factors such as particle size, shape, zeta potential, and structural design [11].

When targeted, rod (needle)-like nanostructures (such as NWs) can affect greater structural damage upon direct contact with cancer cells compared with spherically shaped NPs [11,12]. In addition, one of the reported mechanisms for the inherent anticancer activity of Pt-NPs is via strand breakage in the chromosomal DNA, leading to the inhibition of cell replication and apoptosis [13,14]. Studies have also demonstrated its low cytotoxicity to noncancerous cells such as Peripheral Blood Mononuclear Cells (PBMCs) at concentrations of 200 μg/mL [15]. However, dose-dependent cytotoxicity was reported for prostate cancer cells (LNCaPs), ovarian teratocarcinoma (PA-1), lung adenocarcinoma (A549), pancreatic cancer (Mia-Pa-Ca-2), breast cancer cells (MDA-MB-231), and human neuroblastoma cancer cells (SH-SY5Y) [15,16,17]. Jacob and co-workers also reported that pristine Pt-NPs can cause cancer cell death via apoptosis (SiHa/U251) and that the composite Au-Pt mechanism of cell death (SiHa) occurs via cell shrinkage, chromatin condensation, and intra-nucleosomal DNA fragmentation [18].

A majority of the studies on Pt-based cytotoxicity against cancer cells were performed using NPs [19]. Interestingly, the cytotoxicity specifically of Pt-NWs coupled to devices for neural signal detection has been studied with no significant anticancer cytotoxic effects reported [20,21]. Hence, the effect of the structural design (geometry) of pristine Pt-NWs has not been fully explored, and their cytotoxicity profile cannot be assimilated to conventional Pt-NPs due to geometric factors [22].

Therefore, this study focused on progressing this aspect and aimed to synthesize composite Fe-Pt-NWs to exploit their dual properties as a new form of chemotherapy against germ-cell-derived testicular cancer cells (Tera 1) and fibroblast (3T3) cells to assess their potential for targeted chemotherapy. In addition, the approach to synthesize the Fe-Pt NWs was modulated to control the structural design, magnetic properties, and cyto-activity of the nanosystem, using two approaches, viz Method 1 (elemental Pt-NWs) and Method 2 (a Fe-Pt NW alloy). These variants were compared with pristine Pt-NWs. The magnetic properties of the synthesized NW composites were determined to serve as a selectively active magnetic targeted chemotherapeutic delivery system.

## 2. Materials and Methods

### 2.1. Materials

Anodic Aluminum Oxide (AAO) membranes with a stated diameter of 40 nm were purchased from InRedox (Longmont, CO, USA). Metal salts consisting of Fe (II) sulphate heptahydrate (F8263); Na_2_SO_4_ (238597); hexachloroplatinic acid (262587); and the chemicals boric acid; ascorbic acid; hydrochloric acid, perchloric acid, nitric acid, and sodium hydroxide were procured from Sigma-Aldrich (St. Louis, MO, USA). Tera-1 and 3T3 cells were purchased from Cellonex (Johannesburg, South Africa). McCoy’s 5A, fetal bovine serum, RPMI 1640, and penicillin–streptomycin was purchased from Thermo Fisher Scientific (Waltham, MA, USA).

### 2.2. Chemical Electrodeposition Setup and Synthesis of Pt-NWs and Fe/Pt Composite NWs

All NW platforms were synthesized using a template electrodeposition approach in a conventional three-electrode setup. Firstly, free-standing Anodic Aluminum Oxide (AAO) membrane with an average pore size of 40 ± 4 nm was sputter coated with a 30 nm Au layer, using turbomolecular pumped coater (Quorum Q150T ES) in an argon atmosphere. 

During the electrochemical deposition process, the Au layer acted as the working electrode that enabled pore filling with the metal substrates. A small Pt plate was used as the counter electrode, and an Ag/AgCl (3 M KCl) reference electrode (Metrohm AG Herisau, Switzerland) was employed. To ensure accurate reading potentials were measured, two general practices were followed. Measurements were recorded against the reference electrode, and this electrode was placed at a set distance from the working electrode, minimizing the uncompensated resistance. The electrodes were connected to a potentiostat (PGSTAT302N, Autolab, Utrecht, Netherlands).

The electrolyte used in the deposition of the Pt-NWs was composed of 1% *w*/*v* H_2_PtCl_6_ and 7% *w*/*v* HClO_4_. The Pt-NWs were deposited at a constant voltage of −0.35 V [23]. The magnetic properties of Pt-NWs were altered by the addition of Fe, using two methods. The first Fe/Pt composite NW (Method 1) was synthesized by initially depositing Pt into the AAO template; the pore diameter of the AAO membrane was then increased using the method utilized by Gu and co-workers [6]. The Pt-deposited AAO templates were submerged in 5% H_3_PO_4_ for 60 min at 25 °C. The space created between the Pt NW and AAO pore channels was then deposited with Fe, using pulsed deposition in the following electrolyte: 12% *w*/*v* FeSO_4_ (7H_2_O), 7.5% *w*/*v* Na_2_SO_4_, 2.5% *w*/*v* HBO_3_, and 1% *w*/*v* ascorbic acid. The deposition of the Fe was carried out at −1.1 V (30 s pulses), with sufficient time intervals between pulses (30 s) allowing for the migration of Fe (II) ions into the pore channels. The second composite NW (method 2) was prepared by depositing the NWs in an electrolyte containing both Pt (IV) and Fe (II). The electrolyte used was 12% *w*/*v* FeSO_4_ (7H_2_O), 7.5% *w*/*v* Na_2_SO_4_, 2.5% *w*/*v* H_3_BO_3_, 1% *w*/*v* C_6_H_8_O_6_, 1% *w*/*v* H_2_PtCl_6_, and 7% HClO_4_. The NWs were synthesized using pulsed electrodeposition, alternating between −1.1 V and −0.35 V for 50 s and 5 s, respectively. 

### 2.3. Release of NWs from AAO Membranes

The sputter-coated Au layer was removed using Aqua Regia (3:1), minimizing contact time to prevent the dissolution of the Fe in the NWs. The AAO templates were then dissolved using 1 M NaOH and sonicated to ensure the release of the MNWs. The released MNW were then collected using an external magnet or centrifugation, washed three times, and suspended in ethanol for storage to preserve their magnetic properties [24]. 

### 2.4. Characterization of NWs

#### 2.4.1. Quantification of Fe and Pt in Synthesized NWs

The Fe content in Fe/Pt composite NWs (Methods 1 and 2) and Pt content in Pt-NWs and Fe/Pt composite NWs (Methods 1 and 2) were quantified against an external calibration curve, using inductively coupled plasma–optical emission spectrometry (ICP-OES) (Shimadzu ICPE-9820). The analytical wavelength chosen for the quantification of Fe was 259.94 nm, and 203.646 nm was chosen for the quantification of Pt. All samples were analyzed in triplicates. Sample preparation briefly involved aliquoting 100 µL of well-dispersed NW suspension into a volumetric flask, together with 300 µL Aqua Regia, and then leaving it at 300 K for 60 min for complete dissolution. Prior to analysis, the resulting sample was diluted to 20 mL and analyzed using the ICP-OES. 

#### 2.4.2. Morphological Analysis of Free and AAO Embedded NWs

The TEM sample was prepared by diluting the MNW suspensions with absolute ethanol and then ultrasonicating it for two minutes, using Vibra-Cell^TM^, Sonics^®^ Sonics & Material Inc., (Newtown, CT, USA). A drop of this suspension was placed onto a carbon-coated copper grid, with a filter paper being used to absorb excess sample. The copper grid was left overnight to air-dry. The samples were then analyzed using a TEM (FEI Tecnai T12 TEM) and HRTEM (JEM-2100). 

For analyses of the NWs embedded in the template for pore widening confirmation, a SEM (JEOL JSM-7500F FE-SEM) was used. The NW embedded AAO template was attached to the stub with carbon tape before analysis.

#### 2.4.3. Electrical Stability of the Synthesized NWs

The NWs from the ethanol dispersion were removed from the dispersion by using an external magnet or centrifugation, rinsed three times with deionized water, and then redispersed and diluted with deionized water as well. The resultant NW suspensions were then placed into a consumable cell (DTS1070), and the zeta potential was measured in triplicate, using Malvern Zetasizer Nano (Malvern Instruments, Malvern, UK).

#### 2.4.4. Magnetic Characterization of the Synthesized NWs

A Vibrating sample magnetometer (VSM) (Quantum Design DynaCool, 12 T Physical Property Measurement System (PPMS)) was utilized to quantify the magnetic fields around the synthesized NW platforms. The measurements were taken with the applied field parallel and transverse to the long axis of the NWs embedded in the AAO template. The isotherm magnetic measurements were taken at 300 K by sweeping the magnetic field in the range of ±10,000 Oe.

#### 2.4.5. Analysis of the Outer Oxide Formation on Fe/Pt Composite NWs

The ethanol-suspended NWs were pipetted onto a clean glass microscope slide and allowed to air-dry. Thereafter they were analyzed using a Horiba LabRAM HR Raman spectrometer with an Olympus BX41 microscope attachment. A low laser power of 0.4 mW was used so that the magnetite would not be converted to hematite, as shown by Faria et al. [25]. The 514.5 nm laser wavelength used was from a Lexel Model 95-SHG argon ion laser.

#### 2.4.6. Determination of the Crystalline Structure of the Synthesised NWs

X-ray diffraction patterns were acquired for analyzing crystalline phases employing a MiniFlex 600 benchtop X-ray diffractometer (Rigaku, Tokyo, Japan) with Cu Kα radiation (*λ* = 1.54060 Å). The X-ray tube was setup at a voltage of 40 kV and a current of 15 mA. The region of 30° ≤ 2θ ≤ 90° was evaluated.

#### 2.4.7. Biocompatibility, Cytotoxicity, and Cellular Internalization Analysis

##### Cell Viability (%)

Two cell lines were employed in this study namely, Tera 1 (Germ cell derived Testicular cancer cells) and 3T3 (Mouse Fibroblasts). Tera 1 was obtained from Cellonex (Johannesburg, South Africa) and revived using McCoy’s 5A fortified with 20% fetal bovine serum (FBS) and then incubated at 37 °C and 5% CO_2_. Tera 1 and 3T3 were cultured using RPMI 1640 enriched with 10% FBS and 1% penicillin–streptomycin incubated at 37 °C and 5% CO_2_. The experimental cells were then seeded into a 96-well plate, using the following concentration in each well: Tera 1 (5400 cells/well) and 3T3 (4500 cells/well). The wells were then separated into groups and treated with 10 μL of either positive control (cisplatin); negative control (PBS); or treated with Pt NWs, Fe/Pt composite NWs (Method 1), or Fe/Pt composite NWs (Method 2). The 96-well plates were again incubated at a temperature of 37 °C and 5% CO_2_ environment for 24 and 48 h to assess the cell viability so that short-term biocompatibility or cytotoxicity could be determined. The % viability of the cells was determined using the (sodium 3′-[1-(phenylaminocarbonyl)-3,4-tetrazolium]-bis (4-methoxy6-nitro) benzene sulfonic acid hydrate) (XTT) assay. Spectrophotometric analyses were carried out for each well, using a microplate reader, Victor X3 microplate reader (Perkin Elmer, Waltham, MA, USA). Absorbance values were taken at a wavelength of 450 nm, and the reference wavelength was measured at 690 nm. The % viability was calculated using the following formula:$$Cell\;Viability\;\left(\%\right)=\frac{AV\;of\;treated\;cells-RV\;of\;treated\;cells}{AV\;of\;untreated\;\left(control\right)\;cells-RV\;of\;untreated\;cells}\times 100$$
where *AV* is absorbance value, and *RV* is reference value.

Experiments were performed in triplicate, and statistical significance was ascertained by determining the *p*-value, using an unpaired *t*-test.

##### Cellular Internalization

The cell lines were cultured, seeded, and treated as previously mentioned in a 96-well plate. After 48 h, the cells were washed thrice with PBS to remove the free-floating non-internalized NWs. The interaction between the synthesized NWs and the cells were then investigated and imaged using a phase-contrast microscope (Olympus CKX53).

## 3. Results

### 3.1. NW Fabrication and Characterization

NWs were fabricated using template-assisted electrodeposition of respective solutions of metal salts, using porous alumina membranes. Electrodeposition is a well-reported electrochemical method to grow nanostructures on these templates [26]. These AAO membranes were also sputter coated with gold to serve as a cathode during the electrodeposition process. This template is suitable for this study, as the pore diameters are uniform and can be controlled, resulting in the diameters of the synthesized NWs being controlled [27]. These membranes were used to prepare NWs with a modulated composition and structure.

Two methods were used to synthesize composite NWs. Method 1 used a two-pot, two-step electrodeposition process. The initial step involved the deposition of the Pt core of the NW. Thereafter, the pore size of the AAO template was enlarged, as shown in Figure 1. The second step electrodeposited the outer Fe shell creating a core–shell structure with high control of the dimensions of the core. Method 2 employed a one-pot synthesis, using a single electrodeposition step utilizing a single electrolytic bath consisting of the combined electrolytes used in the Pt and Fe electrodeposition steps in Method 1. This produced alloyed Fe and Pt-NWs that were electrodeposited with the atoms randomly orientated. The two methods were employed in this study to examine the variance in the physical properties, geometric structure, and cytotoxic activity of the NWs produced by the one-pot and two-pot synthesis. 

The average dimensions of the synthesized NWs were obtained using transmission electron microscopy (TEM) and scanning electron microscopy (SEM) images, and they are presented in Table 1. 

The standard error of the mean (SEM and TEM) was used to specify the statistical uncertainty. Pt-NWs had an average length of 2.2 µm, with a standard deviation of 1.2 µm (Min = 0.2 µm/Max = 2.0 µm, over an average of n = 50 measurements). For the Fe/Pt composite NWs, the Fe/Pt composite NWs of Method 1 had an average length of 2.2 µm (SD = 1.0/Min = 0.303 µm/Max = 5.827 µm over an average of n = 50 measurements), and the Fe/Pt composite NWs of Method 2 displayed the largest average length of 3.1 µm (SD = 1.8/Min = 0.557 µm/Max = 8.446 µm over an average of n = 50 measurements). With regards to the diameter, Pt NWs depicted an average diameter of 39 nm whilst Fe/Pt composite NWs (Method 2) produced an average diameter of 41 nm Fe/Pt composite NWs (Method 1) had a significantly larger average diameter of 54 nm when compared to Pt-NWs (39 nm), which was a precursor in the synthesis of Fe/Pt composite NWs (Method 1). This is due to the pore-widening step after the initial Pt deposition in the AAO template which created the space necessary to deposit the Fe in a controlled manner.

The Fiber Pathology Paradigm identified a 10 µm threshold for the length of NWs, above which a ‘frustrated phagocytosis’ phenomenon occurs which can lead to inflammatory responses, which are undesirable for targeted delivery [28]. Ag NWs with smaller diameters (30 nm) have previously shown that they were internalized by cells through phagocytosis, while NWs of larger diameters (100 nm) induced ‘frustrated phagocytosis’ [29]. The nominal diameters (40–50 nm) and small lengths (<3.1 µm) suggest that the synthesized NWs can be internalized by the cells without inducing ‘frustrated phagocytosis’, and this can be beneficial in targeted chemotherapy. 

By synthesizing NWs of elemental Pt and composite Fe/Pt, we planned to study the effects of elemental Pt-NWs in comparison to the Fe/Pt composite NWs, with particular interest in the magnetic and the structural properties of the NWs and their consequence on the potency of the cytotoxic effect of the NWs on fibroblasts and testicular cancer cells. To analyze the structural properties and morphology of the NWs, HRTEM and TEM were used. 

Figure 2A,C,D correspond to the low-magnification HRTEM images of the synthesized NWs. All the NWs have a smooth and consistent morphology. However, the HRTEM image of a Fe/Pt composite NW (Method 2) shows a single NW that had a change in diameter. However, most of the NWs maintain their consistent nominal diameters. The images also show the fragmentation of each of the NWs. This is caused by the sonication step during the releasing process of the NWs from the AAO template. 

As seen in Figure 2, the Fe/Pt composite NWs tend to accumulate in clusters, and observation that is consistent with previous reports of magnetic NWs [30]. This impacted the physical stability of the NW suspensions, as the small NW clusters aggregated and sedimented in a brief period. This effect was reversable, as agitating the NW suspension redispersed the suspension. Although Fe/Pt composite NWs (Method 1) had a significant zeta potential of −34.6 mV, the electrostatic interactions were not great enough to overcome the magnetic and gravitational forces acting on the NWs, leading to a greater sedimentation rate. Pt NWs and Fe/Pt composite NWs (Method 2) had good stability as well, given that they had a zeta potential of −22.3 mV and −24.6 mV, respectively, and smaller magnetic interactions between the NWs. 

Higher magnification images were also studied using HRTEM, corresponding to Figure 2B,D,F. Fe/Pt composite NWs (Method 1) displayed a core–shell structure with an outer amorphous layer surrounding the NWs with an average thickness of 3.9 ± 2.0 nm. The Fe/Pt composite NWs (Method 2) also showed an outer amorphous layer surrounding the NWs with an average thickness of 2.3 ± 0.4 nm that had greater regularity compared to Fe/Pt composite NWs (Method 1). This outer amorphous layer was formed due to the basic NaOH environment that the NWs were exposed to during the releasing process, which causes the outer Fe to oxidize. This oxidation is well established in previous studies of Fe NWs and conforms to the results obtained in this study [31,32]. The outer shell of the Fe/Pt composite NWs (Method 1) was not smooth and continuous as that of the Fe/Pt composite NWs (Method 2). The amorphous layer was very rough and resembled amorphous spots on the NW surface. The synthesized NWs were observed to have a polycrystalline structure, as shown in the high-magnification images in Figure 2, with the planes randomly arranged and non-constant interplanar spacing. The grains are not well defined due to the high electron density of Pt.

The crystal structure of NWs has been previously shown to affect cytotoxicity. Single-crystal Fe NWs have a greater cytotoxicity when compared to polycrystalline Fe NWs, as demonstrated by Ivanov et al. This is likely due to the prevention of intracellular dissolution of the Fe NWs due to the passivation provided by the Fe oxide shell of the Fe NWs [24,33]. It was therefore vital to characterize the crystal structure of the synthesized NWs.

The crystal structure of the synthesized NWs was examined; thereafter, the crystallite size and d-value were estimated by studying the powder XRD patterns of the synthesized NWs. Figure 3 depicts the obtained diffractograms of the synthesized NWs, and Table 2 summarizes the calculated crystallite size (Scherrer equation) and d-value (Bragg’s law). 

Pt NWs were indexed to five main peaks at 2θ values of 39.75°, 46.20°, 67.56°, 81.31°, and 85.90°, corresponding to planes (111) (200), (220), (311), and (222) in a cubic crystal system. Fe/Pt composite NWs (Method 1), which contained both Fe and Pt, showed only five peaks, which correspond to a cubic crystal system for Pt, as was observed for Pt NWs. The observed peaks for Fe/Pt composite NWs (Method 1) were at 2θ values of 39.80°, 46.26°, 67.66°, 81.39°, and 85.9°, corresponding to hkl values of (111), (200), (220), (311), and (222). Fe/Pt composite NWs (Method 2) were deposited in an electrolyte bath containing Pt (IV) and Fe (II), leading to an alloy formation indexed to iron–platinum ((Fe_3_Pt_17_)0.2). The key peaks observed in the data for Fe/Pt composite NWs (Method 2) were at 2θ values of 40.09°, 46.50°, 68.10°, 81.95°, and 86.67°, corresponding to (111), (200), (220), (311), and (222) planes of a cubic crystal system. All the synthesized NWs had minor peaks at ≈77° due to the presence of the Au sputter coated onto the back of the AAO template. 

The Scherrer equation was used to estimate the crystallite sizes in the deposited NWs.
$$D=\frac{K_{s}\lambda }{\beta \cos\theta }$$
where *Ks* is the shape factor, 0.9, as used by Terohid et al. for tungsten oxide nanowires [34]; *λ* is the wavelength of Cu Kα; *β* is the full width at half maxima (FWHM) in radians of the observed peaks; θ is the peak position in radians; and *D* is the crystallite size in nm. The estimated crystallite sizes were smaller than the diameter of Pt NWs (39 nm), Fe/Pt composite NWs (Method 1) (50 nm), and Fe/Pt composite NWs (Method 2) (41 nm), indicating that these NWs are polycrystalline, as also confirmed by the HRTEM. Since individual crystallites have a minimum of one magnetic domain, Pt NWs, Fe/Pt composite NWs (Method 1), and Fe/Pt composite NWs (Method 2) can be considered to have a complex multi-domain state. 

Fe/Pt composite NWs (Method 1) have larger grain sizes compared to Fe/Pt composite NWs (Method 2). The smaller the crystallite sizes, the larger the ratio of grain boundaries which provide greater strength and hardness to the NWs [35]. Since Fe/Pt composite NWs (Method 2) have smaller grain boundaries and, therefore, greater strength and hardness, it would explain why Fe/Pt composite NWs (Method 2) had the largest average length of the NW. The strength of Fe/Pt composite NWs (Method 2) resisted the breakage of the NWs through the sonication step during synthesis. 

An outer oxide layer on the Fe/Pt composite NWs of Method 1 and Fe/Pt composite NWs of Method 2 was developed during the release process of the NWs from the AAO template, as observed by the HRTEM in Figure 2. This layer naturally formed when the Fe from the composite NWs was in the presence of NaOH and air. The oxides formed during this process have implications on the magnetic properties and the attachment potential for coatings and small molecules used in engineered delivery nanoparticle systems for active drug targeting [32]. It was therefore necessary to determine the composition of the outer oxide layer, and thus was accomplished using Raman Spectroscopy (Figure 4).

The Fe/Pt composite NWs (Method 1) fluoresce when using the Ar^+^ 514.5 nm wavelength, and this made it difficult to obtain a well-resolved Raman spectrum of significant SNR. However, peaks at 309 cm^−1^ (E_g_) and 658 cm^−1^ (A_1g_) were observed. Combined with the confirmed presence of O on the EDX analyses (Figure 5) of Fe/Pt composite NWs (Method 1) and a very low intensity peak at 2θ value of 35° from the XRD diffractogram, this indicates that the observed outer oxide layer is likely magnetite as well. Fe/Pt composite NWs (Method 2) presented very low intensity peaks at 310 cm^−1^ (E_g_) and 658 cm^−1^ (A_1g_), suggesting that the outer oxide layer comprised magnetite as well. The low intensity of the peak could have been brought about by the presence of an oxide-layer thickness (2.3 nm) that is below the detection limit of the Raman spectrometer or that is amorphous or has Raman inactive modes. 

### 3.2. Magnetic Characterization of the Synthesized NWs

The efficacy for magnetic targeted delivery is directly proportional to the ability of the NWs to be coupled to an external static or dynamic magnetic field without any adverse thermal effects due to eddy currents. For successful magnetic targeted delivery, the magnetic force on the synthesized NWs will need to overcome the forces present in the capillary system, mainly the viscous drag force, where the NWs will be trapped and removed from the vascular system and magnetically guided to the target area (diseased tissue). Therefore, for NWs to be viable as a magnetic targeted delivery system, they will need to exhibit a significant response to an applied magnetic field. This makes ferromagnetic and superparamagnetic behavior desirable for magnetic targeted delivery platforms [36]. 

Pt NWs were measured with the field both parallel and transverse to the long axis of the NW (Figure 6A). Along the parallel axis, Pt NWs displayed predominantly diamagnetic properties where magnetization decreased with the increasing field strength. Along the transverse axis, Pt NWs showed competing diamagnetic and superparamagnetic properties. 

Figure 6B shows the normalized M-H hysteresis loops of Fe/Pt composite NWs (Method 1) in the parallel and transverse direction to the applied magnetic field. In the parallel orientation, the long field scan ±10,000 Oe (not shown here) showed a combination of competing interactions, i.e., diamagnetism and ferromagnetism. The competition is evident with shorter magnetic field sweeps ±5000 Oe. Figure 6B,C are zoomed in to 5000 Oe to facilitate the viewing of the magnetic behavior at a higher resolution. Soft ferromagnetism is ascribed to the Fe/Pt composite NWs, as illustrated by the hysteresis loop, while the diamagnetism is attributed to the Au-coated AAO template. The diamagnetism decreases the induced magnetization with an increasing applied magnetic field at 300 K. Clear soft magnetic behavior is evident in the hysteresis loop for the parallel orientation for field sweeps in the range of ±4000 Oe. A square ratio of M_R_/M_S_ = 0.62 was observed. For the transverse orientation, a hysteresis loop was also found with a square ratio of M_R_/M_S_ = 0.2. The magnetic easy axis of Fe/Pt composite NWs (Method 1) is therefore in the parallel axis. The values of the magnetic saturation and remanence were deduced for Fe/Pt composite NWs (Method 1) to M_S_ = 7.62 × 10^2^ A/m and M_R_ 4.21 × 10^2^ A/m, respectively, with a coercive field determined to be Hc = 476 Oe for the parallel axis at 300 K. The magnetization of Fe/Pt composite NWs (Method 1) was fully saturated at a low field of 1800 Oe.

Figure 6C shows the normalized M-H loops of Fe/Pt composite NWs (Method 2). The transverse orientation shows paramagnetic behavior. Paramagnetic materials are slightly attracted to a magnetic field. The parallel orientation shows a very noteworthy response, as the M-H loop resembles that of two competing ferromagnetic grains or nanowires of varying dimensions [37]. The combination of the competing interactions makes Fe/Pt composite NWs (Method 2) not suitable for magnetic targeted delivery due to fidelity in the signal-to-noise ratio and response. 

As the goal of this study was to analyze the potential of the synthesized NWs as an agent in magnetic targeted delivery, NWs that would be attracted towards a magnetic field with a sufficient Ms at low H were desired. This would allow the NW platform to be trapped and extracted from the bloodstream and transported to the target site. 

Pt NWs displayed both diamagnetic behavior and superparamagnetic behavior. This competing property does not allow Pt NWs to produce an unequivocal and significant response to an applied magnet field, as the orientation of the NW cannot be controlled when released from the AAO template, thus making Pt NWs unsuitable for magnetic targeted delivery. The use of Pt NW would necessitate a large magnetic field, which may have adverse effects on the human tissue. 

Fe/Pt composite NWs (Method 1) displayed ferromagnetic behavior in the parallel and transverse orientation and, thus, behaved as a soft magnetic material for a polycrystalline ferromagnetic material. The magnetic properties of Pt NWs were, thus, successfully changed to ferromagnetic behavior with the coating of the Pt with Fe that is suitable for magnetic targeted delivery [38].

### 3.3. Cytotoxicity and Cellular Internalization of the Synthesized NWs

#### 3.3.1. Cytotoxicity of the Synthesized NWs on Fibroblast Cells

The biocompatibility of the synthesized NWs was assessed by observing the effect of the synthesized NW platforms on %viability 24 h and 48 h after treatment on non-cancerous fibroblast 3T3 cell cultures, using the XTT assay. The concentrations of Pt NWs, Fe/Pt composite NWs (Method 1), and Fe/Pt composite NWs (Method 2) were 320, 160, 80, 40, and 20 µg/mL. This corresponds to a concentration of ~11,200, ~5600, ~2800, ~1400, and ~700 NWs/cell for Pt NWs; ~6400, ~3200, ~1600, ~800, and ~400 NWs/Cell for Fe/Pt composite NWs (Method 1); and ~7200, ~3600, ~1800, ~900, and ~450 NWs/Cell for Fe/Pt composite NWs (Method 2). These concentrations are much greater compared to the targeted concentrations in potential biomedicine applications. These high concentrations were chosen to investigate the effects of the NW composition on the short-term acute toxicity of the synthesized NWs [39].

Pt (II) complexes such as cisplatin have been widely used for cancer therapeutics with good effect. They do, however, have major drawbacks due to their non-selectivity bringing about severe adverse effects and drug resistance [40]. Pt (0) nanoparticles have also been examined to evaluate their cytotoxicity, and they are suggested to effectuate oxidative stress, which brings about a loss in cell viability [41]. The effect of the change in morphology of Pt (0) nanoparticles towards a NW anisotropy on cytotoxicity was evaluated in this research as well. The cell viability for 3T3 cells treated with Pt NWs showed no significant decrease and stayed at ≥90%, except at the highest concentration of 320 µg/mL at the 24 h time point (Figure 7A). At 48 h, the cells at the same concentration showed significant recovery, with cell viability of >90% (Figure 7B) and cell morphology looking healthy, as shown in Figure 7C, indicating that Pt NWs were not cytotoxic to 3T3 cells at these high concentrations, even though they were clearly internalized by the cells.

Coating the Pt nanowires with Fe (Fe/Pt composite NWs (method 1)) had a significant influence on the cell viability of treated cells, as shown by the results for Fe/Pt composite NWs (Method 1; Figure 7A,B). Twenty-four hours after treatment with the Fe/Pt composite NWs (Method 1), the cell viability significantly decreased at concentrations of 160 and 320 µg/mL. After 48 h, the cells treated with 160 µg/mL of Fe/Pt composite NWs (Method 1) recovered to ≥90%. At a concentration of 320 µg/mL, however, the cell viability stayed below 70%. Fe/Pt composite NWs (Method 1) were also shown to be internalized by the cells (Figure 7D). NW made from an Fe-Pt alloy (Fe/Pt composite NW (Method 2)) showed the greatest cytotoxic activity, as shown by the results for Fe/Pt composite NWs (Method 2), even though the phase-contrast images suggest that they internalized the least number of NWs (Figure 7E). At 24 h and 48 h time intervals, Fe/Pt composite NWs (Method 2) displayed greater toxicity at the higher concentrations of 160 and 320 µg/mL. At concentrations of 80, 40, and 20 µg/mL, for both 24 h and 48 h incubation periods after treatment with Fe/Pt composite NWs (Method 2), there was a negligible decrease in cell viability, indicating that, at these concentrations, Fe/Pt composite NWs (Method 2) were biocompatible on 3T3 cells.

#### 3.3.2. Cytotoxicity of the Synthesized NWs on TC Cells

To assess the cytotoxic activity of the synthesized NWs against testicular cancer, Tera 1 cells treated with the synthesized NW platforms were incubated with varying concentrations for 24 h and 48 h. The concentrations chosen for treatment on Tera 1 cells were the same concentrations as those used for 3T3 to provide comparative results to the cytotoxic activity for the different cell lines. These concentrations expressed as NWs/Cell for Pt NWs: ~8800, ~4400, ~2200, ~1100, and ~550 NWs/Cell; Fe/Pt composite NWs (Method 1): ~5600, ~2800, ~1400, ~700, and ~350 NWs/Cell; and Fe/Pt composite NWs (Method 2): ~6400, ~3200, ~1600, ~800, and ~400 NWs/Cell.

The respective NW platforms clearly demonstrated that they were internalized by the Tera 1 cells (Figure 8C–E) and displayed inherent toxicity against the Tera 1 cell in a dose-dependent manner. Pt NWs revealed significant cytotoxic activity at a high concentration of 320 µg/mL and had a high IC_50_ of 354 µg/mL at 48 h (Figure 8B). Pt NWs also showed a slight cell recovery at 48 h for lower concentrations. Fe/Pt composite NWs (Method 1) were the best performing and had the lowest IC_50_, i.e., 165 µg/mL, at 48 h when compared to the Pt NWs and Fe/Pt composite NWs (Method 2). The toxicity of Fe/Pt composite NWs (Method 1) showed a dose- and time-dependent trend, with the cell viability decreasing with increasing concentrations and incubation time. Fe/Pt composite NWs (Method 2) presented an IC_50_ of 207 µg/mL at 48 h, which is superior to Pt NWs but was inferior when compared to the Fe/Pt composite NWs (Method 1). The cell viability did not appear to be significantly influenced by a longer exposure but showed dose-dependent toxicity. It should be noted that the chosen concentrations tested for both 3T3 and Tera 1 are far greater than the envisioned concentrations in the bloodstream and healthy tissue. However, due to the nature of magnetic targeted delivery, it is anticipated that these concentrations will be reached at the target site.

## 4. Discussion

Pt NWs and Fe/Pt composite NWs, using two methods, were successfully synthesized via template electrodeposition and characterized with controllable dimensions. A novel method of creating a core–shell structure by coating a metallic NW with another metal, using electrodeposition in an AAO template, was described. These methods are cost-effective and scalable. With the successful addition of Fe to the Pt NWs as a coating in Method 1, creating a core–shell structure, we were successfully able to manipulate the magnetic characteristics of the NWs, making them suitable for a magnetic targeted delivery system.

Both Pt NWs and Fe/Pt composite NWs (method 2) have inherent activity against the cancerous cell line Tera 1, while also being biocompatible with 3T3 cells. However, they both demonstrated unfavorable magnetic characteristics for future drug delivery systems. Fe/Pt composite NWs (method 1) displayed desirable ferromagnetic properties with a significant M_S_ of 7.62 × 10^2^ A/m. Fe/Pt composite NWs (method 1) were additionally deemed biocompatible when tested against mouse fibroblasts (3T3). Fe/Pt composite NWs (method 1) presented the most significant cytotoxic activity against Tera-1 cells, with an IC50 of 165 μg/mL at 48 h, which was around twice the potency of Pt NWs. The addition of Fe to the Pt NWs significantly increased the potency and exhibited enhanced cytotoxic activity by decreasing cell proliferation and increasing cell death of the cancer cells when compared to pure Pt NWs. The potential mechanisms of the cytotoxic activity include strand breakage of the chromosomal DNA, leading to apoptosis and cell-cycle arrest. Other possible mechanisms comprise the production of hydroxyl radicals and the inhibition of the cellular metabolic activity [13]. Extensive future studies are required to determine the toxicology of Pt and composite Pt NWs. Artificial intelligence and machine learning are promising avenues for determining the mechanisms of the induced and enhanced toxicology [42]. 

There are two limiting factors of the study: the physical instability of the Fe/Pt composite NWs (Method 1) in solution and the high electron density of Pt. The Fe/Pt composite NWs (Method 1) tended to sediment out of suspension due to magnetic and gravitational forces. This could lead to future dosing errors during future studies. A promising solution would be the addition of a polymeric coating to strengthen the electrostatic interactions. The high electron density of Pt hindered the HRTEM analysis of the crystallographic orientations and grain boundaries of the synthesized NWs. 

The results obtained during this study warrant further investigation into Fe/Pt composite NWs (Method 1) in creating targeted multifunctional therapeutic systems in cancer nanomedicine applications due to the ferromagnetic behavior and enhanced cytotoxic activity. This includes preclinical studies focusing on subjects such as the understanding of the mechanism cell death, the effects of increasing the thickness of the outer Fe layer on the Ms value, the potential of this composite as a contrast agent for MRI, and the effect of applying an alternating magnetic field and near-infrared radiation on the cytotoxicity of Fe/Pt composite NW (Method 1) for a multimodal induction of cell death.

## Figures and Tables

**Figure 1 biomedicines-11-01857-f001:**
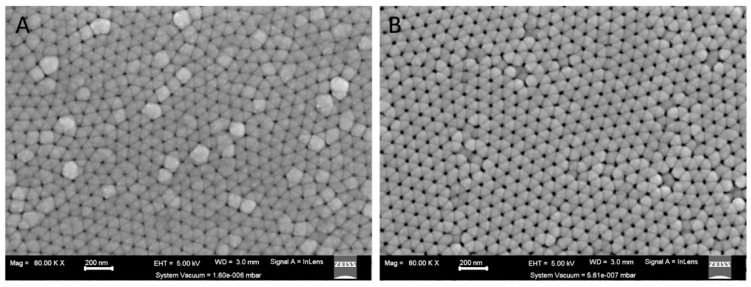
SEM image showing surface view of AAO template (**A**) after Pt deposition and (**B**) after controlled AAO dissolution for nanopore widening where the system vacuum = 1.60 × 10^−6^ mbar and 5.61 × 10^−7^ mbar respectively.

**Figure 2 biomedicines-11-01857-f002:**
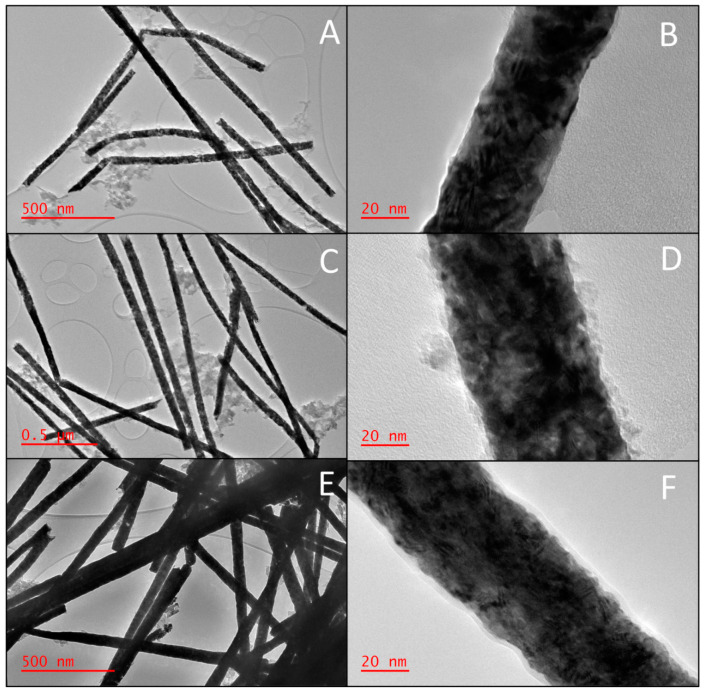
HRTEM images of the NWs depicting the morphology of (**A**) low magnification of Pt NWs, (**B**) high magnification of Pt NWs, (**C**) low magnification of Fe/Pt composite NWs (Method 1), (**D**) high magnification of Fe/Pt composite NWs (Method 1), (**E**) low magnification of Fe/Pt composite NWs (Method 2), and (**F**) high magnification of Fe/Pt composite NWs (Method 2).

**Figure 3 biomedicines-11-01857-f003:**
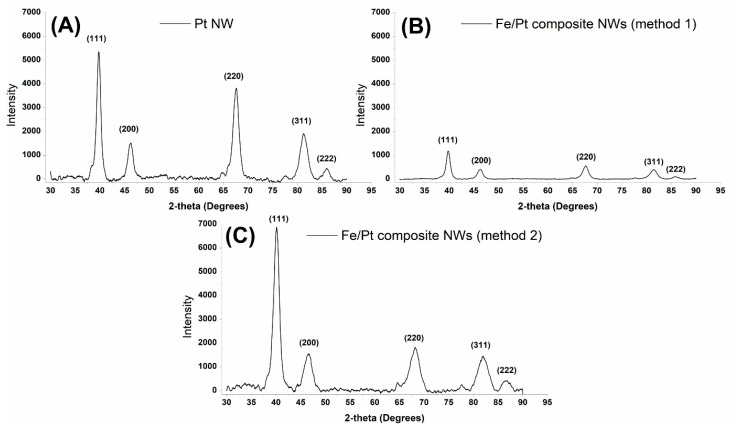
Powder XRD diffractograms obtained for (**A**) Pt NWs, (**B**) Fe/Pt composite NWs (Method 1), and (**C**) Fe/Pt composite NWs (Method 2).

**Figure 4 biomedicines-11-01857-f004:**
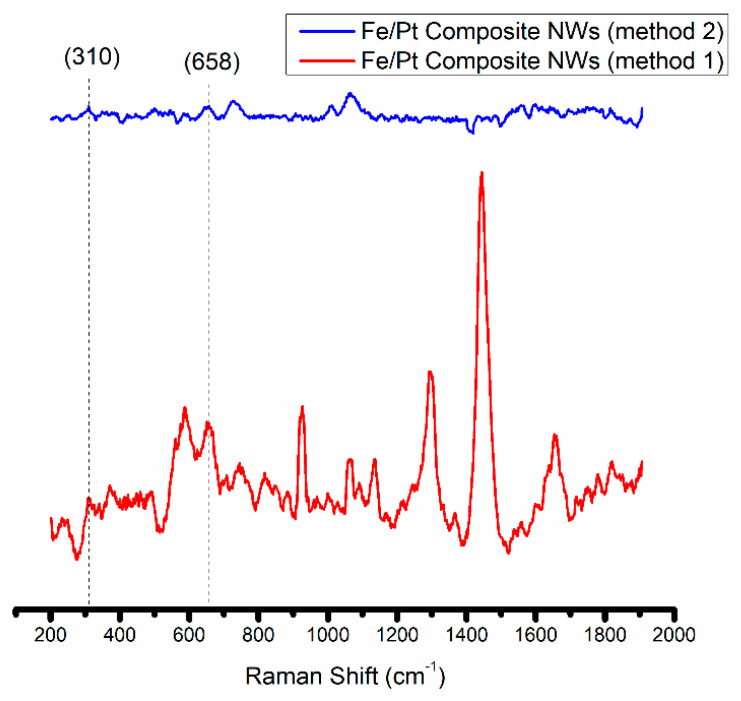
Raman spectra obtained for Fe/Pt composite NWs (Method 1) and Fe/Pt composite NWs (Method 2).

**Figure 5 biomedicines-11-01857-f005:**
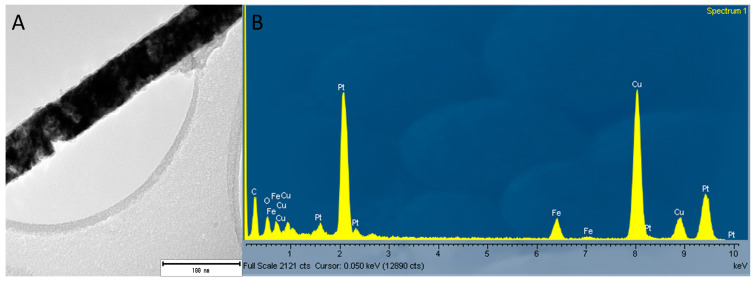
(**A**) Image of a single NW of Fe/Pt composite NWs (Method 1) where the EDX data were obtained and (**B**) EDX analyses of Fe/Pt composite NWs (Method 1) showing the elemental composition of Fe/Pt composite NWs (Method 1) containing Pt, Fe, and O.

**Figure 6 biomedicines-11-01857-f006:**
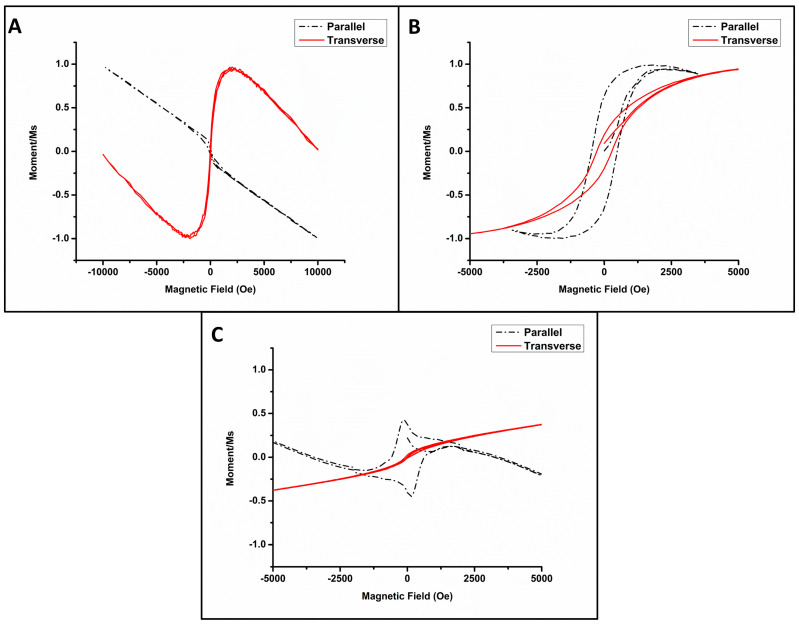
Normalized M − H loops depicting the magnetic behavior of (**A**) Pt NWs, (**B**) Fe/Pt composite NWs (Method 1), and (**C**) Fe/Pt composite NWs (Method 2).

**Figure 7 biomedicines-11-01857-f007:**
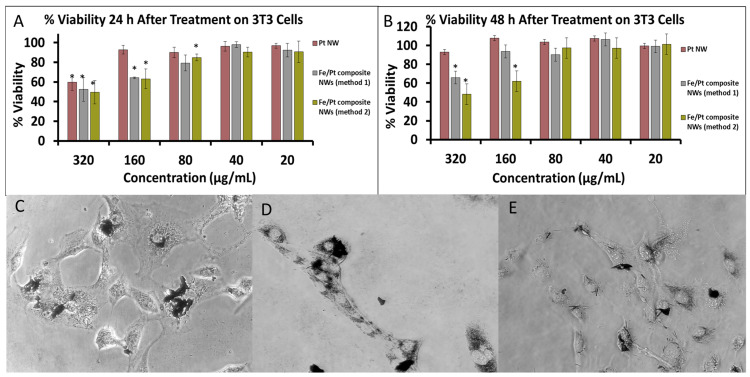
(**A**) %Cell Viability of 3T3 cells after treatment of 24 h. (**B**) %Cell Viability of 3T3 cells after treatment of 48 h. Phase-contrast images of 3T3 cells treated with (**C**) Pt NWs, (**D**) Fe/Pt composite NWs (Method 1), and (**E**) Fe/Pt composite NWs (Method 2) (* denote *p* < 0.05 when compared to untreated cells).

**Figure 8 biomedicines-11-01857-f008:**
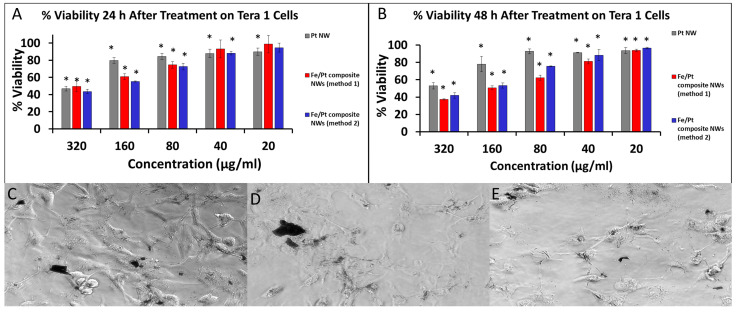
(**A**) %Cell Viability of Tera 1 cells after treatment for 24 h. (**B**) %Cell Viability of Tera 1 cells after treatment for 48 h. Phase-contrast images of 3T3 cells treated with (**C**) Pt NWs, (**D**) Fe/Pt composite NWs (Method 1), and (**E**) Fe/Pt composite NWs (Method 2) (* denote *p* < 0.05 when compared to untreated cells).

**Table 1 biomedicines-11-01857-t001:** Parameters of synthesized NW.

NW Type	Average Length (µm) (n ≥ 100)	Average Diameter (nm) (n ≥ 50)	Average Oxide Layer Thickness (nm) (n ≥ 50)	Zeta-Potential in Deionized Water (mV)
Pt NWs	2.2 ± 1.2	39 ± 2.0	-	−22.3 ± 1.1
Fe/Pt composite NWs (Method 1)	2.2 ± 1.0	54 ± 4.0	3.9 ± 2.0	−34.6 ± 1.1
Fe/Pt composite NWs (Method 2)	3.1 ± 1.8	41 ± 3.0	2.3 ± 0.4	−24.6 ± 1.7

**Table 2 biomedicines-11-01857-t002:** Summary of obtained XRD data and interpretation using Scherrer equation for crystallite size and Bragg’s law for d-value/spacing.

NW Platform	Composition	2θ	Β (FWHM)	Crystallite Size (nm)	d (Å)	h k l
Pt NWs	Pt	39.7	1.05578	7.08	2.27	111
	Pt	46.2	1.33413	5.48	1.967	200
	Pt	67.6	1.49182	4.43	1.39	220
	Pt	81.3	1.82394	3.31	1.183	311
	Pt	85.9	1.33386	4.36	1.13	222
Fe/Pt composite NWs (Method 1)	Pt	39.8	1.06482	7.02	2.26	111
	Pt	46.3	1.38333	5.28	1.96	200
	Pt	67.7	1.2606	5.23	1.38	220
	Pt	81.4	1.83388	3.28	1.18	311
	Pt	85.9	1.47306	3.95	1.13	222
Fe/Pt composite NWs (Method 2)	(Fe_3_Pt_17_)_0.2_	40.1	1.46095	5.11	2.25	111
	(Fe_3_Pt_17_)_0.2_	46.5	2.03405	3.59	1.95	200
	(Fe_3_Pt_17_)_0.2_	68.1	2.42422	2.72	1.38	220
	(Fe_3_Pt_17_)_0.2_	81.9	2.47224	2.43	1.17	311
	(Fe_3_Pt_17_)_0.2_	86.7	1.79009	3.23	1.12	222

## Data Availability

The data presented in this study are available upon request from the corresponding author.

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
