# Peer review of "Design and Evaluation of Composite Magnetic Iron–Platinum Nanowires for Targeted Cancer Nanomedicine"

_biomedicines, 2023, doi:10.3390/biomedicines11071857_

Round 1
Reviewer 1 Report
The research paper titled "Design and Evaluation of Composite Magnetic Iron-Platinum Nanowires for Targeted Cancer Nanomedicine" focuses on synthesizing and evaluating the properties of core-shell platinum and iron-platinum composite nanowires for potential applications in targeted cancer therapy. The study demonstrates enhanced cytotoxic activity and ferromagnetic behavior of the composite nanowires compared to pure platinum nanowires, providing a promising foundation for future magnetically targeted chemotherapeutic approaches. The manuscript would benefit from improvements in the clarity of the objective, methods description, and result presentation. Though, scientific analysis is performed in detail and data in results are presented well, there are some major concerns that needs to be addressed. My comments are given below-
1. Clarify the research objective: The abstract should explicitly state the primary objective of the study, which is to investigate the influence of geometrical structure, magnetism, and cytotoxic activity of core-shell platinum and iron-platinum composite nanowires for potential application in targeted cancer nanomedicine, however, in vitro testing alone is insufficient potential claim in cancer nanomedicine. I suggest modify little. .
2. Elaborate on the synthesis methods: Provide more detailed descriptions of the template electrodeposition techniques used to synthesize the Pt-NWs and Fe/Pt composite NWs. Include information on the specific parameters, such as voltage, current density, and deposition time, to enable reproducibility.
3. Explain the rationale behind using two different synthesis methods: Justify the choice of employing two different methods for synthesizing the Fe/Pt composite NWs. What advantages or benefits does each method offer in terms of control over the core diameter and other relevant properties? Also, add catalogue number of materials, reagents and instruments model number, supplier etc. to make study repeatable in other laboratories.
4. Provide a comprehensive characterization analysis supported with up to date references. While HRTEM, Raman spectroscopy, XRD, ICP-OES, VSM, and zeta potential analysis were performed, more details should be included regarding the specific techniques, instrument models, and experimental conditions employed for each characterization method. Add a latest report https://doi.org/10.3390/ijms24076153 with sentence ´This template is suitable for this study as the pore diameter are uniform and can be controlled resulting in the diameters of the synthesized NWs being controlled’ to mark reference list up to date.
5. Discuss the observed morphological features: Elaborate on the significance of the nominal diameters of 40-50 nm and lengths of approximately <4 μm for the polycrystalline Pt-based NWs. How do these dimensions influence the properties and potential applications of the nanowires?
6. Clarify the polycrystalline structure of the Fe/Pt composite NWs: Explain the implications of the observed polycrystalline structure and provide more detailed information about the crystallographic orientations and grain boundaries within the composite nanowires.
7. Expand the discussion on cytotoxicity assessment: Provide additional details regarding the experimental protocol used for the XTT assay to assess cell viability. Include information on the concentrations of NWs tested, exposure duration, and statistical analysis methods employed.
8. Discuss the mechanism behind enhanced cytotoxic activity citing a recent report https://doi.org/10.1016/j.biopha.2023.114784 on the topic as discussion section I find quite weak. Elaborate on the possible mechanisms that contribute to the enhanced cytotoxic activity exhibited by the Fe/Pt composite NWs compared to Pt-NWs. Is the improved cytotoxicity solely attributed to the ferromagnetic behavior or are there other factors involved?
9. Address potential limitations and briefly discuss implications for future research. Discuss any limitations or challenges encountered during the study, such as possible cytotoxic effects of the NWs on normal cells or potential toxicity related to the iron content. Also, mention any limitations in the experimental design or methods that might affect the validity of the results. Conclude the abstract with a paragraph highlighting the significance and potential impact of the findings. Discuss how the results of this study can contribute to the development of magnetically targeted chemotherapeutic applications and suggest possible directions for future investigations in this field.
Author Response
Kindly see attached.

Reviewer 2 Report
Overall, the central ideas regarding platinum and iron-platinum nanowires for targeted chemotherapy are interesting. I have provided a few suggestions that can improve the quality of this paper;
Abstract
· The level of technical detail is excessive and distracting. Critical information like the nanowire sizes, cell lines used, and key findings are buried in complex sentences.
· The writing lacks coherence and flow. There are abrupt transitions between different characterization techniques and results.
Introduction
· The claims made about platinum and iron-platinum nanowires require stronger evidence. For example, the assertion that "rod (needle)-like nanostructures (such as NWs) can affect greater structural damage upon direct contact with cancer cells compared with spherically shaped NPs" is not supported.
· Some statements are overly broad and ambiguous. For instance, it is not clear what is meant by "alternative strategies are required to alleviate [cancer] limitations." More specificity is needed.
Martials and methods
· Some key specifics are still lacking, such as concentrations of reagents used in nanowire synthesis, electrolyte compositions, electrodeposition voltages and times, and cell seeding densities for cytotoxicity assays. Without this information, the experiments cannot be accurately reproduced.
· Some claimed results are premature and lack sufficient substantiation. For example, the formation of an "oxide shell layer" on the Fe/Pt NWs is asserted but no Raman spectroscopy data are yet presented to demonstrate this.
· Some key specifics are still lacking, such as concentrations of reagents used in nanowire synthesis, electrolyte compositions, electrodeposition voltages and times, and cell seeding densities for cytotoxicity assays. Without this information, the experiments cannot be accurately reproduced.
· Important controls are missing from the cytotoxicity experiments. For example, it is not stated whether untreated cells were tested as negative controls or what positive controls (e.g. cisplatin) were used. Without proper controls, the results cannot be interpreted reliably.
Result
· Reduce the use of colloquial language and try to use more precise terminology. For example, instead of "Interestingly" provide an evidence-based explanation for the results.
· Avoid overly broad comparisons and generalizations. Provide specific comparisons between the different nanowire platforms based on the actual measured parameters (dimensions, crystallite size, magnetic properties, cytotoxicity, etc).
Discussion
· Reducing repetition by consolidating related content. Much of the first two paragraphs is repeated in the third paragraph.
· Being more critical and analytical in interpreting the results. Avoid simply restating the findings and conclusions. Provide deeper insights and relate the results to relevant literature.
· Evaluating the limitations of the current study and providing suggestions for improvements. This will strengthen the credibility and rigor of the discussion.
Revising imprecise language and grammatical errors
Author Response
Kindly see attached

Round 2
Reviewer 1 Report
Accept
Minor editing of English language required